# Temperature Cycle Reliability Analysis of an FBAR Filter-Bonded Ceramic Package

**DOI:** 10.3390/mi14112132

**Published:** 2023-11-20

**Authors:** Wenchao Tian, Wenbin Li, Shuaiqi Zhang, Liming Zhou, Heng Wang

**Affiliations:** 1School of Electro-Mechanical Engineering, Xidian University, Xi’an 710000, China; wenbinli99@stu.xidian.edu.cn (W.L.); 22041212704@stu.xidian.edu.cn (S.Z.); 20041211992@stu.xidian.edu.cn (H.W.); 2Yangjie Electronic Technology Co., Ltd., Yangzhou 225000, China; liming.zhou@21yangjie.com

**Keywords:** FBAR filter, bonded ceramic package, temperature cycle, life prediction

## Abstract

On the background that the operating frequency of electronic devices tends to the radio frequency (RF) segment, a film bulk acoustic resonator (FBAR) filter is widely used in communication and military fields because of its advantages of high upper frequency, ample power capacity, small size, and low cost. However, the complex and harsh working environment puts higher requirements for packaging FBAR filters. Based on the Anand constitutive equation, the stress–strain response of the bonded ceramic package was studied by the finite element method for the FBAR filter-bonded ceramic package, and the thermal fatigue life of the device was predicted. We developed solder models with various spillage morphologies based on the random generation technique to examine the impact of spillage on device temperature reliability. The following are the primary conclusions: (1) Solder undergoes periodic deformation, stress, and strain changes throughout the cycle. (2) The corner of the contact surface between the chip and the solder layer has the largest stress at the end of the cycle, measuring 19.377 MPa. (3) The Engelmaier model predicts that the gadget will have a thermal fatigue life of 1928.67 h. (4) Expanding the layered solder area caused by any solder overflow mode may shorten the device’s thermal fatigue life. The thermal fatigue life of a completely spilled solder is higher than that of a partially spilled solder.

## 1. Introduction

With the popularization of 5G communication, the development of filter technology has received more and more attention. The filter is one of the essential components of communication equipment. Its main function is to screen the signal’s frequency and reserve the required frequency band in the signal to improve the transmitted signal’s anti-interference performance and signal-to-noise ratio [1]. At present, surface acoustic wave (SAW) filters and film bulk acoustic resonator (FBAR) filters are widely used in the field of communication [2]. SAW filters work less well than FBAR filters in the high-frequency region because of the restrictions on upper frequency (2.5–3 GHz), structure, material, operating mode, quality factor, tolerance, and other factors [1]. The upper frequency of the FBAR filter is as high as 20 GHz, the power capacity is large, the size is small, and the cost is low [3]. FBAR filters are regarded as the ideal GHz band filter devices for a future in which electronic equipment operates at high-frequency bands [4]. The thermal stress produced inside the device under an alternating temperature load may result in chip cracking, solder junction cracking, bottom filler delamination, binder delamination, and other problems since different materials have varied physical properties. Therefore, domestic and international research has focused on the device’s temperature reliability [5,6]. In 2014, Yong Xin Zhu et al. conducted temperature tests on lead-free solder at maximum temperatures of 75 °C and 125 °C and predicted the fatigue life of solder with a Coffin– Manson model. The results showed that with the increase in cycles, the maximum stress decreased gradually, and the fatigue cracks generally appeared at the solder corner and expanded along the strain concentration area [7]. Guruprasad Arakere of Intel Corporation of the United States studied the performance of FCBGA under temperature cycle load, and the research results showed that the solder joint failure was related to the load size and the PCB board thickness, and the solder joint at the corner was more likely to fail [8]. In 2018, Li S et al. reported that the package design significantly impacted the device’s reliability by simulating the temperature cycle effect with reciprocating mechanical loading and unloading and tracking the cracking of the chip bond resin by monitoring the sensor’s resistance. Proper package design can improve device reliability despite the inevitability of thermal–mechanical stresses. It comprises packaging material selection, rational component placement, geometric structural optimization, and local design optimization in high-stress areas [9]. In 2021, Dpiver Joshua A. et al. used ANSYS 19.0 software to establish BGA solder joint models of different solders. They used four different life prediction models of Coffin–Manson, Engelmaier, Solomon, and Syed to predict their thermal fatigue life, respectively [10]. A new life prediction model based on damage parameter combination is proposed here. In 2022, Watanabe Yusuke et al. developed a new device for measuring fatigue crack growth and studied the effects of temperature cycle conditions and filler silica content on fatigue crack growth. The results showed that the probability of crack initiation increased with the increase in temperature. The fatigue crack under thermal load propagated faster than that under mechanical load. Adding lower filler increased the crack propagation resistance and improved the thermal fatigue life of electronic devices [11]. In 2022, Watanabe Yusuke et al. developed a new device for measuring fatigue crack growth. They investigated the effects of temperature cycling conditions and silica content in the lower filler on fatigue crack growth. The results showed that the probability of crack initiation increased with the increase in the temperature cycle temperature. Fatigue cracks under thermal load propagated faster than those under mechanical load. Adding lower filler increased the crack propagation resistance and increased the thermal fatigue life of electronic devices [12].

Presently, the research methods used by many scholars mainly include the reliability testing method and the finite element numerical analysis method; the results obtained by the two methods have a slight difference. In the case of limited experiments, many scholars use the finite element method to study the reliability of devices. The need for FBAR filters is gradually growing with electrical technology advancements, and the service environment is becoming more demanding. The research on the reliability of small packages of filters is still in the initial stage. In addition, due to the extrusion of the chip during the package molding process, the solder in the middle layer will inevitably overflow, thus reducing the solder’s smoothness and affecting the entire device’s reliability. However, there are few studies on the impact of solder overflow on the device’s reliability. Therefore, this paper studied the mechanical behavior of a typical FBAR filter under temperature cycling load using temperature-transient structures on bonded ceramic package filters using ANSYS 19.2, and predicted the thermal fatigue life of the filter based on the stress–strain results. Then, using the random generation algorithm, 12 random solder overflow models were generated, which were divided into two types, complete overflow and incomplete overflow, to predict their thermal fatigue life, analyze the influence of solder overflow on the temperature cycle reliability of the device, and finally, to prove that the thermal fatigue life of complete overflow solder is better than that of incomplete overflow solder by a permutation test.

## 2. Temperature Cycle Reliability Analysis and Life Prediction of Devices

### 2.1. The Establishment of the Bonded Ceramic Package Filter Finite Element Model

Figure 1 shows the three-dimensional model of the bonded ceramic package filter established by Solid works; the explosion diagram is shown in Figure 2. It can be seen from the explosion diagram that the bonded ceramic package device is composed of a cover plate, welding frame, ceramic substrate, chip, solder, and pin. Among these, the material of the solder is 92.5Pb5Sn2.5Ag, and the cover plate, welding frame, and pin are all Covar alloys. In addition to the cover plate, the pin, welding frame, and ceramic substrate are tightly sintered together by ceramic co-firing technology.

Due to the small size of the bonded wire, it has little influence on the thermal fatigue life of the device. In order to improve the quality of the finite element mesh and the analysis efficiency and save calculation time and resources, the bonded wire was ignored during the thermal structure coupling simulation. Figure 3 shows the finite element model.

### 2.2. Material Parameters and Meshing

Table 1 shows the material characteristics, including the density, Young’s modulus, Poisson’s ratio and the coefficient of thermal expansion (CTE), thermal conductivity, and the specific heat capacity [13,14]. In addition to the solder, the elastic modulus and Poisson’s ratio of other materials did not change with the temperature between −55 °C and 125 °C, so it was set to a fixed value. Solder is a viscoplastic material. When the ambient temperature is more than half of the melting point (183 °C), it will produce inelastic deformation, and the elastic modulus and Poisson’s ratio will change with the temperature change. Table 2 shows the specific values [14]. The Anand model was used to characterize the solder’s time-independent plastic deformation and time-dependent creep deformation. The parameters of the Anand viscoplastic solder model are shown in Table 3 [14].

The structure of the model analyzed in this paper was relatively simple and had no large curvature boundary. In order to ensure the calculation accuracy, the second-order hexahedral mesh was used for grid division. The total number of elements was 35,763, the number of nodes was 188,703 and the average element quality was 0.95015. All components had at least two layers of mesh, and the mesh mass of the solder part was close to 1. Figure 4 shows the partitioning results.

### 2.3. Temperature Load and Boundary Conditions

In the process of finite element simulation, in order to avoid the phenomenon of excessive thermal expansion constraint, three points displacement constraints method was used to limit the 6 degrees of freedom of the model with the least constraint to prevent rigid body motion, as shown in Figure 5. Point A limited the displacement in the X, Y, and Z directions, point B limited the displacement in the Y and Z directions, and point C limited the displacement in the Y direction. Figure 6 shows the temperature loading curve. The initial temperature of the model was 20 °C; the cycle temperature was −55~125 °C; the high and low temperature holding time was 600 s; and the heating (cooling) conversion time was 60 s. In order to ensure accuracy and occupy fewer computing resources, the temperature load of five cycles was selected for this simulation study, and ten substeps were set for load, with a total solving time of 6660 s.

### 2.4. Temperature Cycle Result Analysis and Life Prediction

Figure 7 shows the 200× magnification of the warping deformation of the model at 5580 s (low-temperature hold) and 6240 s (high-temperature hold) during the temperature cycle. As seen from Figure 7a, at 5580 s, the package had a concave posture. The solder showed an inward contraction state, the deformation near the edge of the chip was large, and the maximum deformation was 0.00208 mm. It can be seen from Figure 7b that at 6240 s, the package had an upward arch posture. The solder expanded outward, and the deformation near the chip side and edge was large, with a maximum deformation of 0.00269 mm.

Figure 8 shows the equivalent stress distribution for the bonded ceramic package device 6660 s. It can be seen from Figure 8a that the stress field outside the entire device was evenly distributed, and the value was small. Figure 8b shows that the stress on the contact surface between the chip and the solder layer was enormous, and the maximum equivalent stress was located at the edge of the chip with a value of 19.377 MPa. It can be seen from Figure 8c that the stress distribution between the solder layer and the chip contact surface was relatively uniform, and the stress at the edge position was the largest. The value was 18.93 MPa. After five temperature cycle cycles, each material had different thermal expansion coefficients, and under the constraint of the displacement conditions, a large stress was generated inside the package; the stress reached the maximum at the corner point where the chip and the solder were connected. Therefore, it can be analyzed and concluded that in the temperature cycle experiment, the solder at the edge of the chip cracks first and leads to delamination, which is consistent with the experimental results of Chang et al. [15].

Figure 9 shows the curve of the maximum equivalent stress of the solder layer over time during the temperature cycle. During the temperature cycle process, the maximum equivalent stress of the solder layer reached 35.10 MPa. According to the fourth strength theory of materials, the maximum equivalent stress of the solder has exceeded the yield strength, and the solder will exhibit viscoplastic characteristics [16]. It can be seen from Figure 9 that the stress of the solder varied periodically with the loading of the temperature load. In the heating stage, the solder stress decreased and fluctuated slightly. In the cooling stage, the solder stress increased greatly. In the low-temperature holding stage, the stress was maximum; in the high-temperature holding stage, the stress was minimum.

The above phenomenon occurs because during the heating process, the increase in temperature causes the softening of solder, and the increase in solder plasticity leads to decreased stress accompanied by fluctuations. During the cooling process, the decrease in temperature causes the solder to harden, and the increase in brittleness causes the stress to rise sharply. The solder is in a relatively stable environment in the high and low temperature holding stages. The local relaxation process gradually releases the stress accumulated inside the solder, and the stress is slightly reduced. In the cooling stage, more stress is accumulated inside the solder, so the stress drop is larger in the low-temperature holding stage.

According to the bonded ceramic package devices’ deformation and stress analysis, thermal fatigue cracks will first appear at the solder edge. The maximum node of solder equivalent stress is defined as the dangerous point; the shear plastic strain of the dangerous point was extracted. By comparing the shear plastic strain amplitudes in XY, XZ, and YZ directions, it was found that the strain amplitudes in the XY direction were the largest. So, the strain amplitudes in this direction play a decisive role in the failure of solder. Figure 10 shows the XY shear plastic strain curve of the solder danger point, and Figure 11 shows the danger node’s shear stress and shear strain hysteresis. It can be seen from the figure that the stress and strain changed periodically with the thermal cycle temperature load; the strain amplitude reached stability after the third cycle, the creep response converged about the fourth cycle, and the shear plastic strain range of the last cycle was 1.75 × 10^−2^.

Based on the statistical results of the thermal fatigue life of 63Sn37Pb solder joints, Engelmaier proposed a thermal fatigue life prediction model of solder based on the Coffin–Mason model and considering the effects of load average temperature and cycle frequency on thermal fatigue life [17,18]. The following formula shows the mathematical equation [19]:(1)$$N_{f}=\frac{1}{2}{\left(\frac{\Delta \gamma_{p}}{2\varepsilon_{f}}\right)}^{1/c}$$
where $N_{f}$ represents thermal fatigue life, $\Delta \gamma_{p}$ represents shear strain amplitude after stabilization, $\varepsilon_{f}$ represents fatigue toughness coefficient, and c represents fatigue toughness index. c is calculated as follows:(2)$$c=-0.442-6\times 10^{-4}T_{m}+1.74\times 10^{-2}\ln(1+f)$$
where $T_{m}$ is the average temperature of the temperature cycle, and f is the cycle frequency in a day. According to the temperature cycle loading conditions and simulation data, $T_{m}$ is 35 °C, c is −0.39, and the fatigue life of the solder temperature cycle is 1928.67 h.

## 3. Influence of Solder Overflow on Temperature Fatigue Life

### 3.1. Establishment of the Overflow Solder Model

Due to the limitations of welding technology, solder overflow occurs when the filter chip is in the bonding process. The bonded solder melts and flows around the chip [20]. Solder will produce local stress concentration after cooling, which is more prone to delamination failure under temperature cycling conditions, affecting the thermal fatigue life of the device. Therefore, based on the Section 2, the influence of solder overflow on the thermal fatigue life of bonded ceramic packaging devices is studied in this section. In order to make the solder overflow morphology in the bonded ceramic package more realistic, we chose to build a random-shaped solder model and conducted a temperature cycle simulation to study the influence of spilled solder on the thermal fatigue life of the filter. The fixed and random-shaped area was generated by a random algorithm of a connected region, surface area, and the thickness of the fixed and random-shaped model as the solder overflow model. The following assumptions were made before modeling:(1)The direction of the solder overflow is entirely random;(2)If we use the same quality of solder to bond the chip, the volume of spilled solder is the same;(3)The solder density is uniform after bonding, without holes, residual stress, and strain.

This paper uses Python’s NumPy, random, and matplotlib modules to generate regions with the same area (1.6 mm^2^) and random shapes. The algorithm implementation process is as follows:(1)As shown in Figure 12, the solder spillable area is first decomposed into 18 × 18 squares.

(2)Select the location of the chip as the initial solder area. Figure 13a shows that the yellow area is the initial solder area. The boundary pool is set up by taking the grid with a blank area in the connected position as the boundary. Figure 13b shows that the green square is the boundary pool, starting from the boundary pool and expanding outwards.

(3)Take a random square from the boundary pool. Suppose its coordinates are (x, y). Find the coordinates of the four points connected to it, namely (x + 1, y), (x − 1, y), (x, y + 1), and (x, y − 1), and determine whether the four points have been filled. Take the points that have not been filled to form an array. In the above array, randomly select a point to fill it. After filling, judge whether all four squares connected to the filling square are filled. If not, the coordinate is added to the boundary pool. Iterate over the squares in the boundary pool, determine whether the four positions connected to the squares have been filled, and remove them from the boundary pool if they have been filled. The specific process is shown in Figure 14. Repeat the above steps N times (N = overflow area/chip area × 100).

(4)Return to step (2) to continue the loop and return the filled result at the end. Figure 15 is a random graph with an area equal to 1.6 mm^2^.

When the boundary pool was expanded during initialization, it could generate a complete overflow pattern. The incomplete overflow morphology could be generated by limiting unilateral boundaries. After getting the overflow morphology, it carried on the smooth processing, as shown in Figure 16.

The finite element model of the solder overflow device was established based on the overflow morphology. The bonded ceramic package filter that hides the cover plate, welding frame, and part of the ceramic substrate is shown in Figure 17.

### 3.2. Thermal Fatigue Life of the Overflow Solder Model

Six models with complete overflow around the perimeter (samples 1–6) and six models with incomplete overflow (samples 7–12) were established, respectively, for temperature cycle simulation analysis, and the solder stress results were obtained, as shown in Figure 18. As seen in Figure 18, when the solder spilled, local regional stress concentration occurred due to the complex morphology of the solder layer. In 6660 s, except for the maximum stress of sample 5, which was 18.915 Mpa, the maximum equivalent stress of the solder layer of all the other samples was higher than that of the non-spillage solder (18.93 Mpa), and the location of the maximum stress was not fixed. The maximum stress area of the complete overflow solder occurred at the edge of the solder layer and the interface of the chip or the edge of the solder layer. The maximum area of incomplete overflow solder stress was more likely on the contact surface between the solder layer and the ceramic substrate. It was mainly located at the edge of the solder layer. From this analysis, it is concluded that spillage of solder will lead to layered expansion.

The Engelmaier model was used to calculate the thermal fatigue life of the samples. The life results of the 12 groups of samples are shown in Table 4 and Figure 19. As seen from Table 4, compared with the thermal fatigue life of the non-spillage solder (1928.67 h), the thermal fatigue life of the FBAR filter will be seriously shortened due to solder spillage. As can be seen from Table 4, the sample mean values of complete overflow and incomplete overflow were 913.36 h and 411.86 h, respectively. Based on the sample data analysis alone, the average life of a solder with full overflow was greater than that with incomplete overflow. However, it is impossible to judge whether the two groups of samples belong to the same distribution by the histogram, and it is also impossible to judge whether the distribution of the two groups of data was independent only of the current calculation results. In order to verify that the thermal fatigue life of complete overflow is better than that of non-complete overflow, a permutation test was needed for the 12 samples.

### 3.3. The Permutation Test

The permutation test was proposed by Fisher in the 1930s as a statistical inference method based on many calculations and using the entire (or random) permutation of sample data [21]. Because it has no requirement on population distribution, it is widely used, especially for small sample data with unknown population distribution and for some hypothesis testing problems where the data are difficult to analyze with conventional methods [22]. The specific process of the permutation test is as follows.

(1)Proposing an original hypothesis H_0_: both groups of samples are taken from the same distribution;(2)Calculate the test statistic: calculate the difference between the means of the two groups of samples, denoted as t_0_;(3)Put all the samples in the same array, then sort them randomly and divide them into two groups (for example, if group A has a samples and group B has b samples, the number of data in the two groups is a and b respectively), and calculate its statistic (the difference between the means of the two groups);(4)Repeat step (3) n times (small sample n is usually 1000) to obtain a series of statistics (t_1_…t_n_);(5)Finally, sort (t_1_…t_n_) from smallest to largest to form (t_1_…t_n_) sampling distribution, and calculate the P (the number of (t_1_…t_n_) greater than t_0_/n). The null hypothesis is rejected if the P is less than the significance level α (α is always 0.05 in statistics).

We mixed the two sets of data, randomly divided into two groups, A1:652.40, 767.83, 719.96, 1289.93, 493.41, 158.32. B1:492.54, 667.47, 1657.84, 130.48, 572.31, 348.82. The difference between the mean values was 35.398. After repeating this process 1000 times, we obtained 1000 data (t_1_…t_n_). The data were made into a frequency table, as shown in Table 5. In total, 33 out of 1000 data were greater than the test statistic t0 (501.50 h).

The probability density chart was drawn according to the frequency table in Figure 20. Green represents the part that is less than the test statistic, and orange represents the part that is greater than the test statistic. The red curve in the figure was the fitted probability density curve, and its area surrounded by the horizontal axis is 1. The area of the orange bar and the horizontal axis was 0.033. The results showed that the probability of occurrence (501.50 h) was only 0.033 under the condition of the original hypothesis (no difference between the effect of complete overflow and incomplete overflow on the temperature cycle reliability).

In conclusion, in the case of the original hypothesis, there was only a 0.033 possibility of 501.50 h or less than 501.50 h average data. α was 0.05 (>0.033), so the original hypothesis was overruled. That is, the thermal fatigue life of the completely solder overflow was higher than that of the incomplete solder overflow [20]. Therefore, incomplete solder overflow should be avoided when bonding FBAR filter chips.

## 4. Conclusions

By using finite element software for a bonding ceramic packing device temperature-transient structure coupling simulation, this paper analyzed the temperature cycle load under the mechanical behavior of an FBAR filter and predicted the filter’s thermal fatigue life according to the result of stress and strain. In addition, different finite element models of solder overflow were established to analyze the influence of solder overflow on the thermal fatigue life of filter chips. The following are the main conclusions:(1)The deformation of the device changes periodically with the loading of temperature load. In the low-temperature holding stage, the whole package presents a downward arch posture, and the whole solder presents an inward shrinking state. In the high-temperature holding stage, the package as a whole is arching, and the solder as a whole is expanding outward. In the whole process, the deformation of the device in the high-temperature holding stage is the largest, and the largest deformation position is near the sharp corner of the chip edge.(2)After five temperature cycles, the stress distribution between the solder layer and the inner side of the contact surface of the chip is uniform, and the stress at the edge is the largest, so it is concluded that the solder at the corner of the chip cracks first. In the heating stage, the solder stress decreases and fluctuates slightly. In the cooling stage, the solder stress increases greatly. The solder stress in the low-temperature holding stage is greater than that in the high-temperature holding stage.(3)The thermal fatigue life of solder is 1928.67 h using the Engelmaier model. When the solder spillage occurs, the solder layer will produce local stress concentration, resulting in layered solder expansion, which seriously affects the thermal fatigue life of the device. The mean thermal fatigue life of the device is 913.36 h and 411.86 h, respectively, when the solder completely overflows and incompletely overflows. The permutation test shows that the device has higher reliability under the temperature cycle when the solder is completely spilled.

## Figures and Tables

**Figure 1 micromachines-14-02132-f001:**
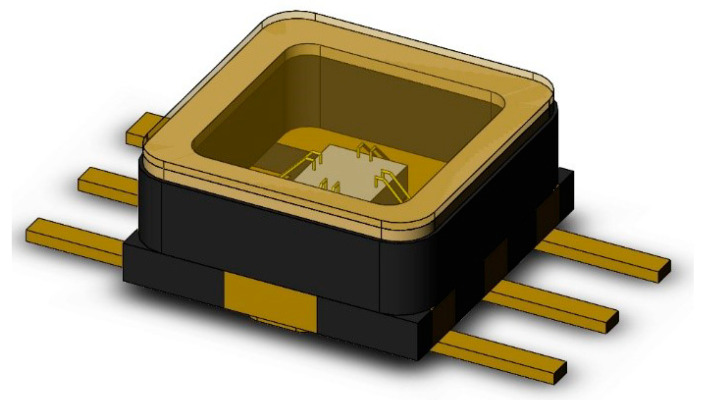
Three-dimensional model of the bonded ceramic package device.

**Figure 2 micromachines-14-02132-f002:**
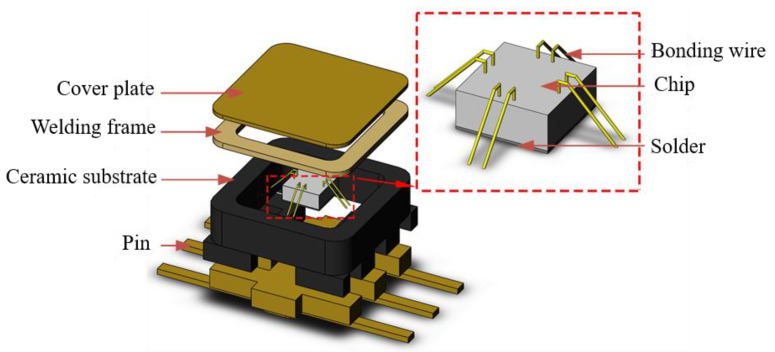
Bonded ceramic package device explosion diagram.

**Figure 3 micromachines-14-02132-f003:**
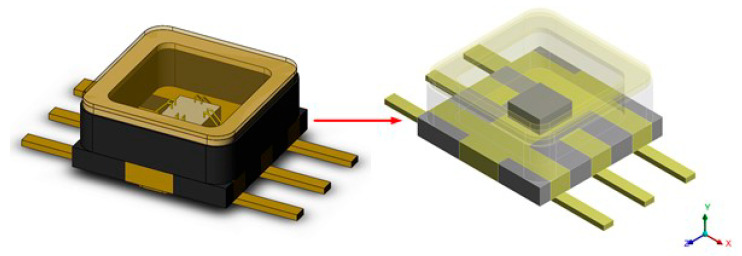
Finite element model of the bonded ceramic package.

**Figure 4 micromachines-14-02132-f004:**
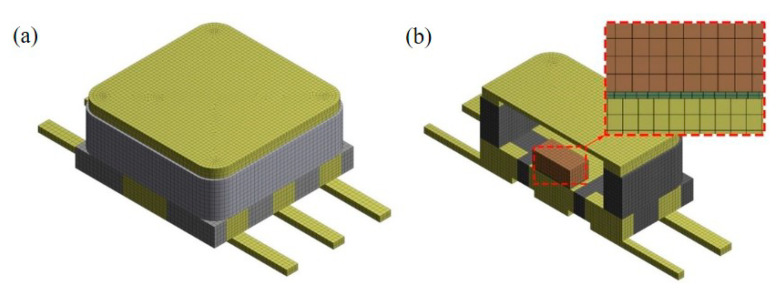
Meshing: (**a**) external meshing, (**b**) internal meshing.

**Figure 5 micromachines-14-02132-f005:**
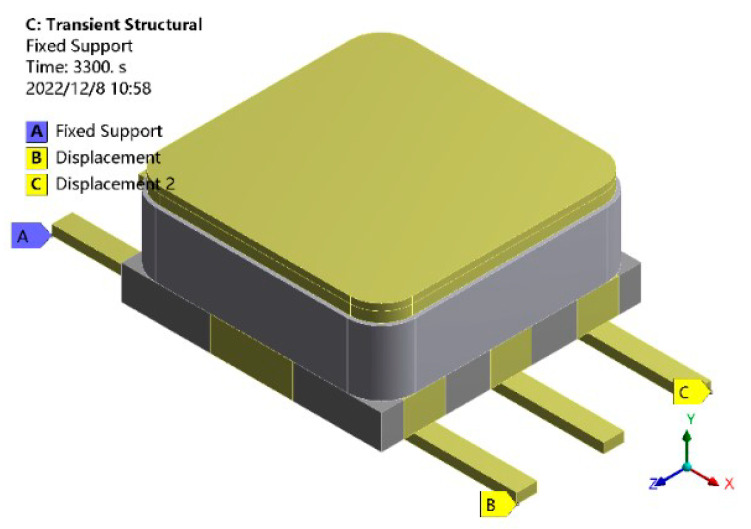
Three-point constraint mode.

**Figure 6 micromachines-14-02132-f006:**
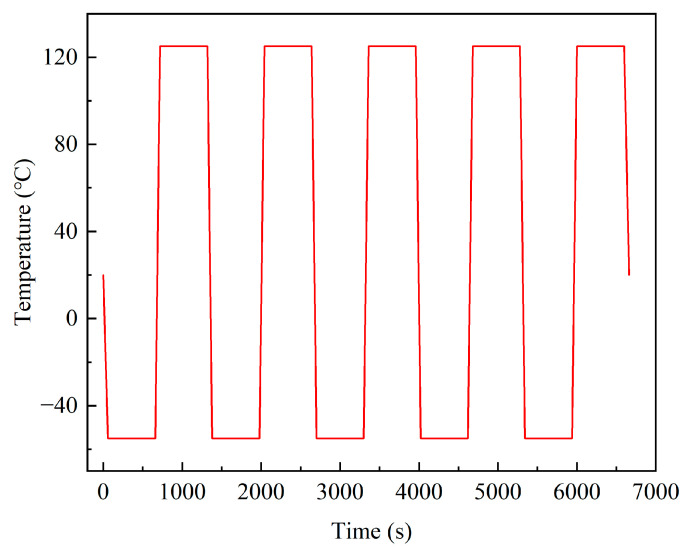
Temperature cyclic loading curve.

**Figure 7 micromachines-14-02132-f007:**
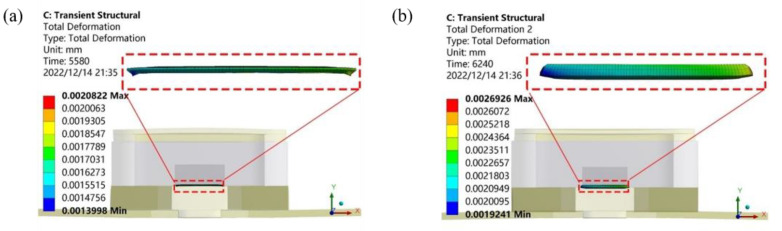
Temperature cycle deformation diagram: (**a**) low-temperature stage deformation, (**b**) high-temperature stage deformation.

**Figure 8 micromachines-14-02132-f008:**
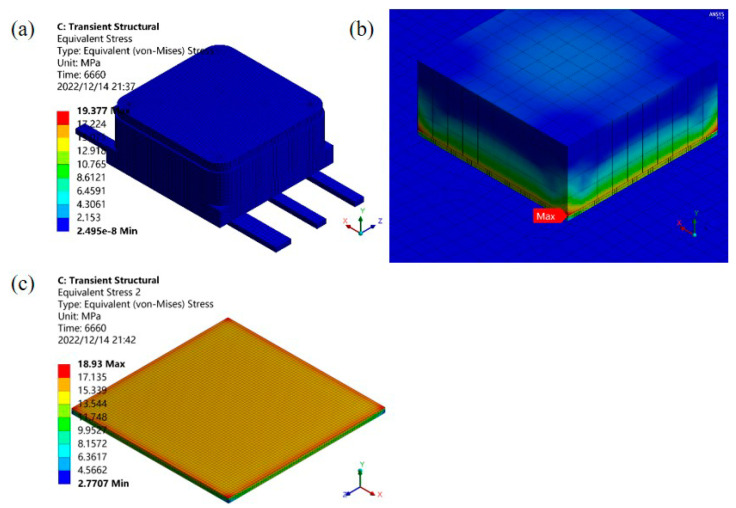
Temperature cycle equivalent stress diagram of bonded ceramic package device: (**a**) equivalent stress diagram outside the model, (**b**) equivalent stress diagram inside the model, (**c**) solder equivalent stress diagram.

**Figure 9 micromachines-14-02132-f009:**
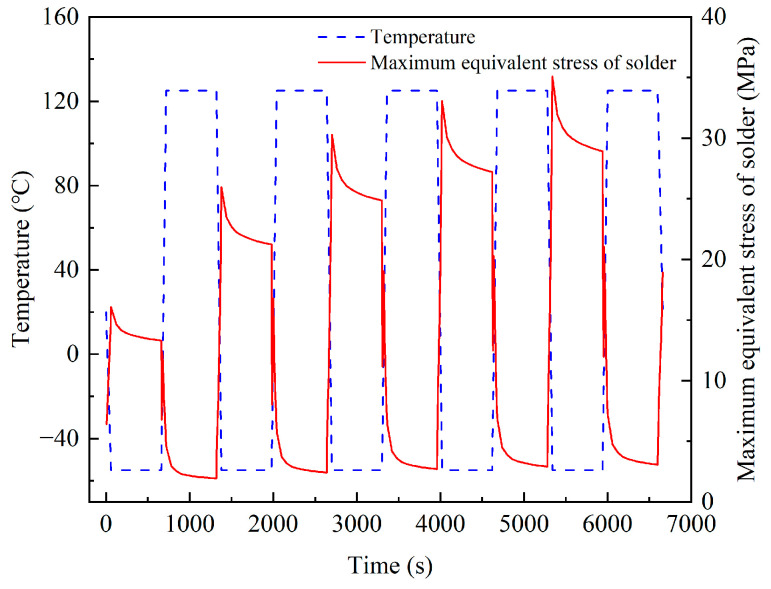
Maximum equivalent stress curve of the solder layer.

**Figure 10 micromachines-14-02132-f010:**
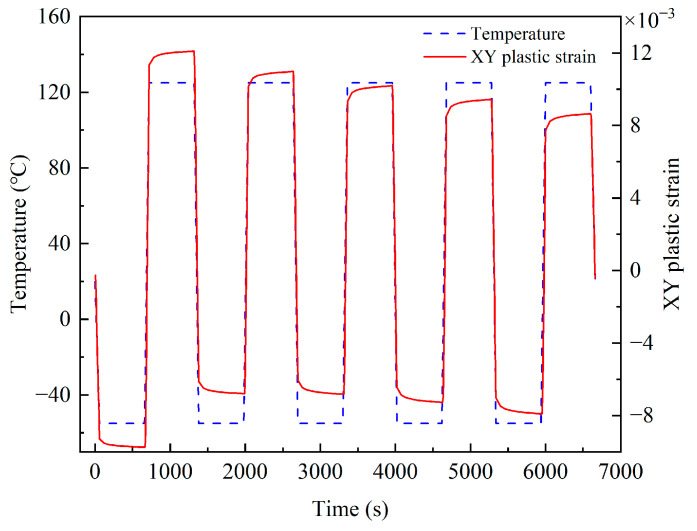
Curve of plastic strain variation in XY direction at the dangerous node.

**Figure 11 micromachines-14-02132-f011:**
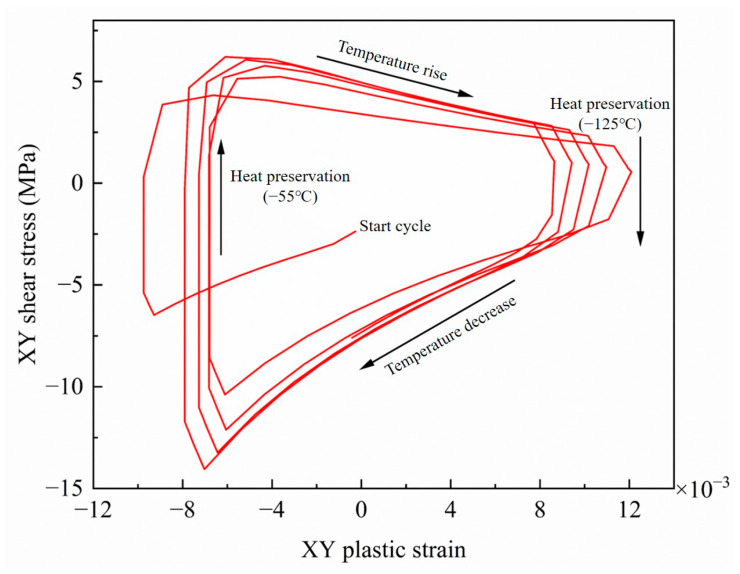
Shear stress–strain hysteresis of the dangerous node.

**Figure 12 micromachines-14-02132-f012:**
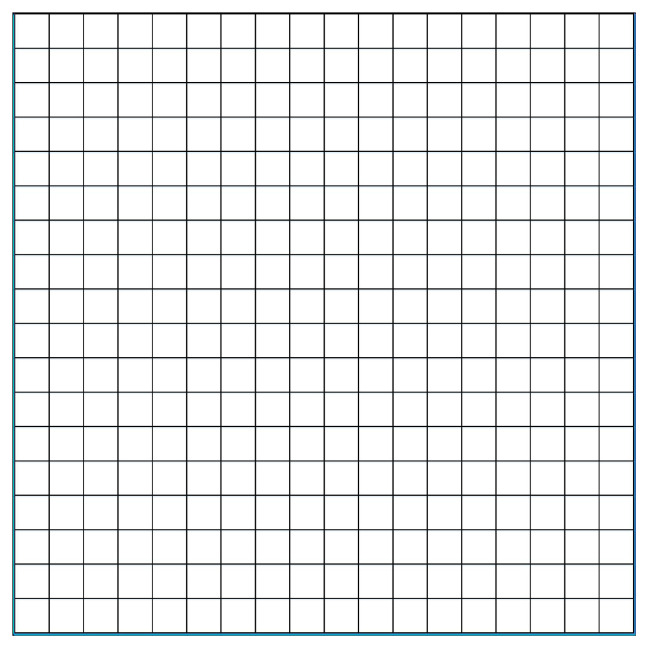
Initialize area.

**Figure 13 micromachines-14-02132-f013:**
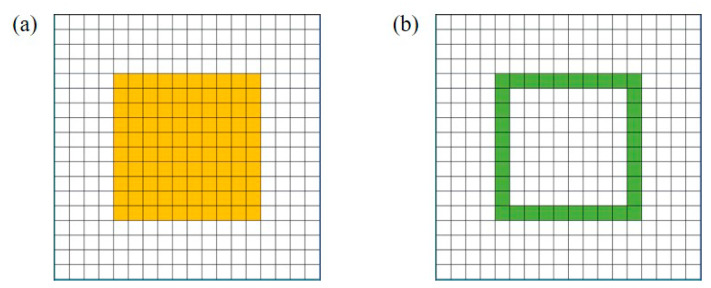
Initialize the solder and boundary pool: (**a**) initial solder area, (**b**) initial boundary pool.

**Figure 14 micromachines-14-02132-f014:**
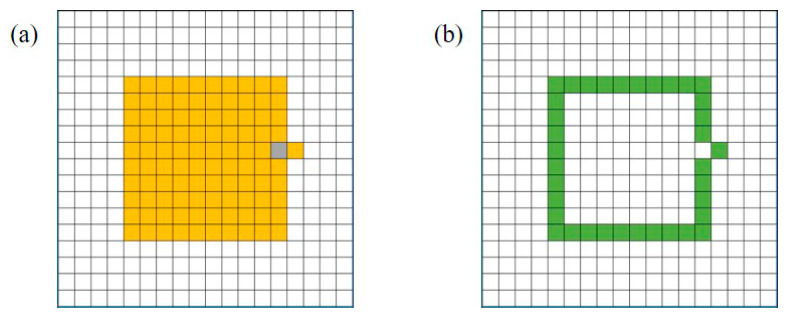
Random update: (**a**) random filling, (**b**) update boundary pool.

**Figure 15 micromachines-14-02132-f015:**
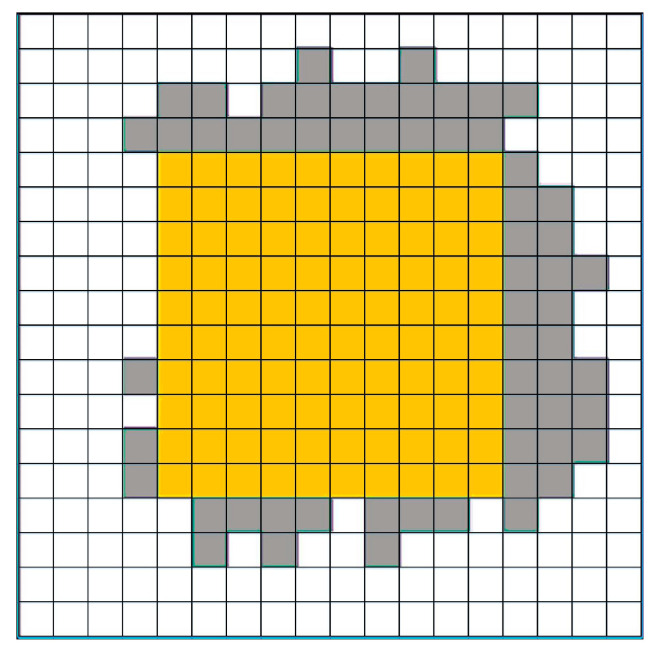
Overflow pattern.

**Figure 16 micromachines-14-02132-f016:**
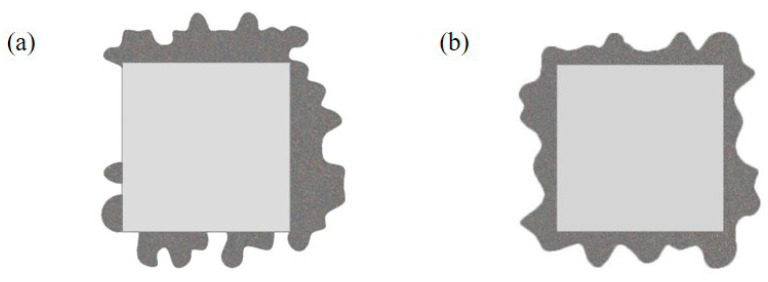
Overflow morphology: (**a**) incomplete overflow morphology, (**b**) complete overflow morphology.

**Figure 17 micromachines-14-02132-f017:**
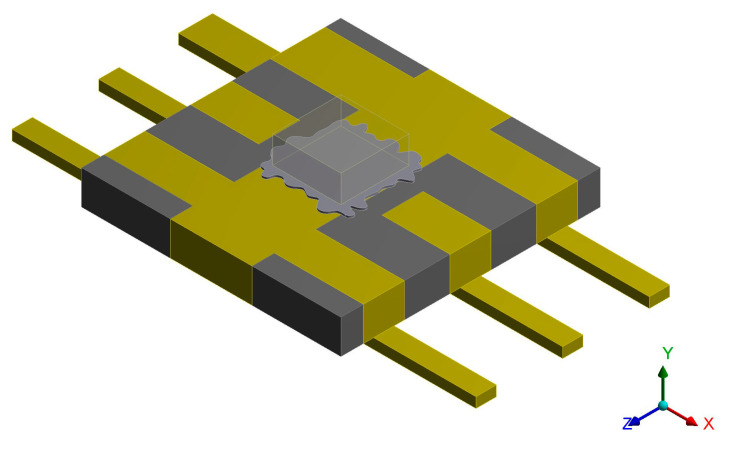
Solder overflow finite element model.

**Figure 18 micromachines-14-02132-f018:**
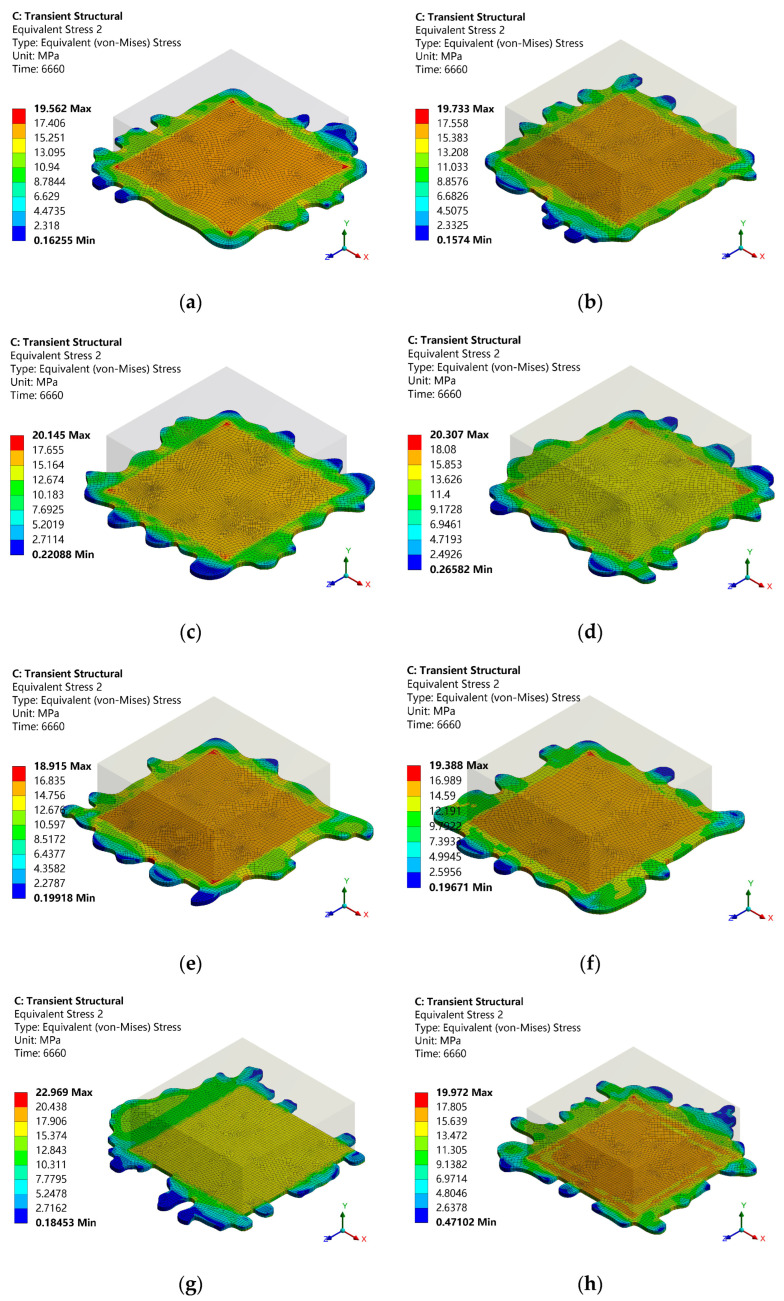
Stress cloud diagram of spilled solder: (**a**) Sample 1, (**b**) Sample 2, (**c**) Sample 3, (**d**) Sample 4, (**e**) Sample 5, (**f**) Sample 6, (**g**) Sample 7, (**h**) Sample 8, (**i**) Sample 9, (**j**) Sample 10, (**k**) Sample 11, (**l**) Sample 12.

**Figure 19 micromachines-14-02132-f019:**
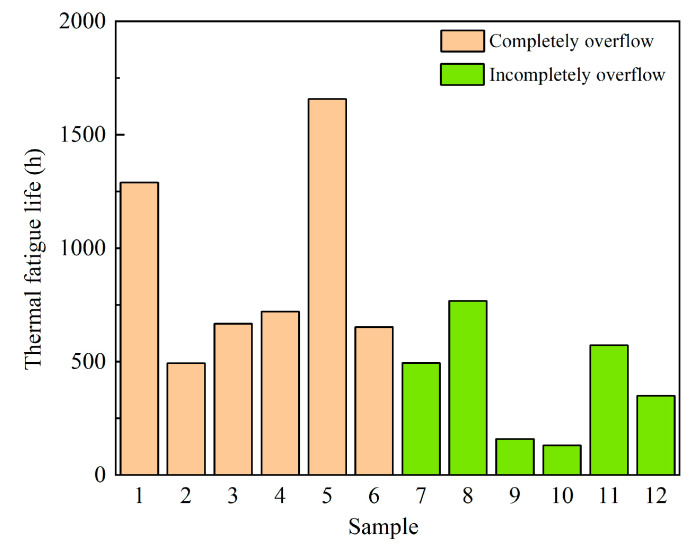
Life histogram of solder under different overflow conditions.

**Figure 20 micromachines-14-02132-f020:**
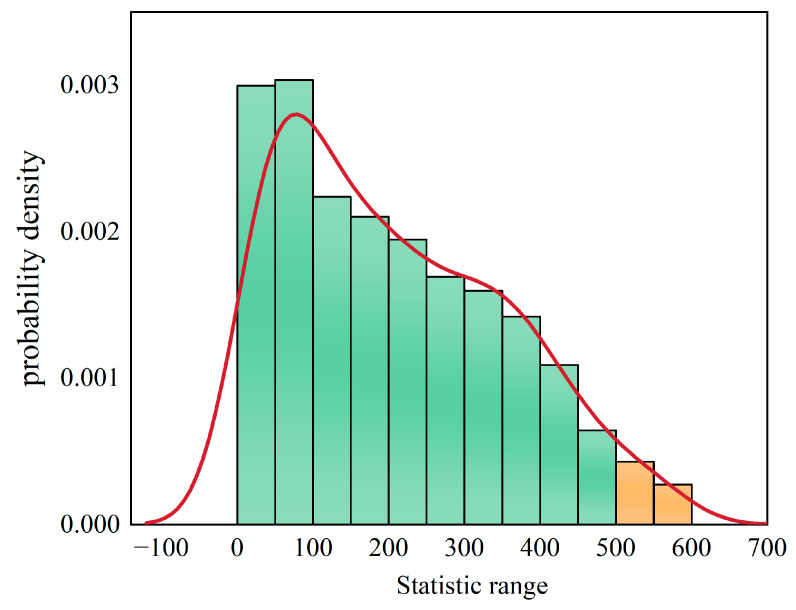
Permutation test probability density.

**Table 1 micromachines-14-02132-t001:** Structural material parameters.

Materials	Density ρ (g/cm^3^)	Young’s Modulus E (MPa)	Poisson’s Ratio ν	CTE α (1 × 10^−6^/K)	Thermal Conductivity λ (W/m·K)	Specific Heat Capacity c (J/(kg·°C))
Kovar	8359	208,000	0.317	5.1	17.3	220
Si	2350	159,000	0.25	2.6	149	702
Ceramic	3100	28,400	0.24	7.1	31	766
92.5Pb5Sn2.5Ag	1075	Table 2	Table 2	29	25	130

**Table 2 micromachines-14-02132-t002:** 92.5Pb5Sn2.5Ag solder elastic modulus.

Temperature (°C)	−55	−35	−15	5	50	100	125
Young’s modulus E (MPa)	25,470	24,930	24,400	23,870	22,670	21,330	20,660
Poisson’s Ratio ν	0.392	0.394	0.397	0.4	0.415	0.427	0.431

**Table 3 micromachines-14-02132-t003:** 92.5Pb5Sn2.5Ag solder Anand viscoplastic model parameters.

Parameters	Value	Definition
s_0_ (MPa)	15.09	Initial value of deformation resistance
Q/R (1/K)	15,583	Activation energy
A (1/s)	3.25 × 10^12^	Pre-exponential factor
ξ	7	Stress multiplier
m	0.143	Strain rate sensitivity of stress
h_0_ (MPa)	1787	Hardening coefficient
Ŝ (MPa)	72.73	Coefficient for deformation resistance Saturation value
n	0.00428	Strain rate sensitivity of saturation value
α	3.73	Strain rate sensitivity of hardening coefficient

**Table 4 micromachines-14-02132-t004:** Lifetime results of different overflow morphology.

Completely Overflow	Lifetime (h)	Incompletely Overflow	Lifetime (h)
Sample 1	1289.93	Sample 7	493.41
Sample 2	492.54	Sample 8	767.83
Sample 3	667.47	Sample 9	158.32
Sample 4	719.96	Sample 10	130.48
Sample 5	1657.84	Sample 11	572.31
Sample 6	652.40	Sample 12	348.82
Mean value	913.36	Mean value	411.86

**Table 5 micromachines-14-02132-t005:** Statistical frequency table.

Statistic Range	Frequency	Statistic Range	Frequency
0–50	154	300–350	82
50–100	156	350–400	73
100–150	115	400–450	56
150–200	108	450–500	33
200–250	100	500–550	22
250–300	87	550–600	14

## Data Availability

Data are contained within the article.

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
