# Peer review of "Temperature Cycle Reliability Analysis of an FBAR Filter-Bonded Ceramic Package"

_micromachines, 2023, doi:10.3390/mi14112132_

Round 1

Reviewer 1 Report

Comments and Suggestions for Authors

In the state of the art, different studies are mentioned on the thermal stress produced inside the devices, but not on its relationship with the size and the material of the package. This is because it is not clear if the novelty of the article is about the reliability of small packages (line 78) of ceramic.

I suggest adding a few sentences between lines 72 and 73 in order to make it clear what the contribution of the research work is.

How were the values ​​shown in tables 1, 2 and 3 decided? I suggest including references that validate the values ​​of each of the parameters. Figures 5 and 6 are not mentioned in the manuscript.

The authors do not discuss whether the results obtained with ceramic package are better o worst compared to other packages. Furthermore, according to the simulations: What is proposed so that the deformation of the device decreases? What is proposed to reduce the effects due to temperature cycles?

Reviewer 2 Report

Comments and Suggestions for Authors

This manuscript discusses the use of simulation to investigate the reliability of a FBAR filter bonded ceramic package. Overall, the manuscript is well-written with sufficient information for reproducing the work. The figures and tables are nicely presented. There are some minor issues as listed in specific comments below.

1. In FIgure 20 and Table 5, what do "statistical range" and "distribution range" represent? can the authors be more specific?

2. How did the authors validate the simulation results? were there any experimental results obtained to cross validate the simulation results?

3. Figure 7 and 8, there's no unit for the time.

Comments on the Quality of English Language

Nil.
